
# Investigation of a Saharan dust plume in Western Europe by remote sensing and transport modelling

Hengheng Zhang[1], Frank Wagner[1,2], Harald Saathoff[1], Heike Vogel[1], Gholam Ali Hoshyaripour[1], Vanessa Bachmann[2], Jochen Förstner[2], Thomas Leisner[1]

[1]Institute of Meteorology and Climate Research, Karlsruhe Institute of Technology, Eggenstein-Leopoldshafen, 76344, Germany

[2]Deutscher Wetterdienst (DWD), Frankfurter Str. 135, 63067 Offenbach am Main, Germany

*Correspondence to*: Hengheng Zhang (hengheng.zhang@kit.edu)

**Abstract.** The evolution and the properties of a Saharan dust plume were studied near the city of Karlsruhe in south-west Germany (8.4298 °E, 49.0953 °N) from April 7 to 9, 2018 combining a scanning lidar (90°, 30°), a vertically pointing lidar (90°), a sun photometer, and the transport model ICOsahedral Nonhydrostatic model - Aerosols and Reactive Trace gases (ICON-ART). The lidar measurements show that the dust particles had backscatter coefficients of $0.86 \pm 0.14$ Mm$^{-1}$ sr$^{-1}$, an extinction coefficient of $40 \pm 0.8$ Mm$^{-1}$, a lidar ratio of $46 \pm 5$ sr, and a particle depolarization ratio of $0.33 \pm 0.07$. These values are in good agreement with those obtained in previous studies of Saharan dust plumes in Western Europe. Compared to the remote sensing measurements, the model simulation predicts the plume arrival time, its layer height, and structure very well but overestimates the backscatter coefficient. In this manuscript, we discuss the complementarity and advantages of the different measurement methods as well as model simulations to predict Saharan dust plumes. Main conclusions are that the ICON-ART model can predict the structure of Saharan dust plumes very well but overestimates the backscatter coefficients by a factor of $2.2 \pm 0.16$ at 355 nm and underestimates the aerosol optical depth (AOD) by a factor of $1.5 \pm 0.11$ at 340 nm for this Saharan dust plume event. Employing a scanning aerosol lidar allows determining backscatter coefficient, particle depolarization ratio and especially lidar ratio of Saharan dust both for daytime and nighttime independently. Combining lidar with sun photometer data allows constraining aerosol optical depth in different ways and determining column integrated lidar ratios. These comprehensive datasets allow for a better understanding of Saharan dust plumes in Western Europe.

## 1 Introduction

Atmospheric dust provides significant impact on the Earth's climate system but the impacts remain highly uncertain (Stocker et al., 2013). These uncertainties are attributed to the larger spatial-temporal variability of aerosol dust and its complex interaction with atmosphere constituents, radiation, and clouds (Satheesh et al., 2006; Min et al., 2008). Besides, dust particles can participate in cloud formation as cloud condensation nuclei (CCN) and these clouds can redistribute solar radiation (Ansmann et al., 2008). Furthermore, dust plumes can modify cloud microphysics and may even change precipitation


distributions (Min et al., 2009; Karydis et al., 2017; Su et al., 2008). Hence, simultaneous observation of clouds and dust plumes can help unravelling the details of dust-cloud interaction processes.

Understanding the distribution and the properties of dust is the key to quantify radiative forcing (Pérez et al., 2006). Since decades, satellites (e.g. Meteosat, Terra & Aqua, CALIPSO) are used to study the properties and transportation of dust around the globe. However, their data still has limitations especially concerning characterisation of the vertical structure of dust

plumes. Intensive field campaigns such as the Saharan Mineral Dust Experiment (SAMUM), conducted in 2006 and 2008, investigated the relation between chemical composition, shape, morphology, size distribution, and optical effects of dust particles with emphasis on vertical profiling of dust optical properties. (Freudenthaler et al., 2009; Müller et al., 2010; Ansmann et al., 2011; Heintzenberg, 2009; Petzold et al., 2009; Kandler et al.,2009). From these field campaigns, the chemical/mineralogical product, microphysics characteristics and optical properties of Saharan dust were studied. The

chemical composition is beyond the scope of this paper. Here we briefly summarize the latter two characteristics. On the African continent, particles with diameters significantly larger than 10 μm were observed e.g. during the SAMUM-1 study. However, in 80% of the cases, the measured particle diameters were below 40 μm. However, in SAMUM-2, the mean dust particle diameter was less than that in SAMUM-1 (Weinzierl et al., 2009; Kandler et al., 2009; Kandler et al., 2011). The authors also find that Saharan dust particles observed during SAMUM–1 and SAMUM–2 were almost non-hygroscopic

(Schladitz et al., 2011). In addition, the complex refractive index of pure dust, the single scattering albedo (SSA) at different wavelengths and the Angstrom exponents (AE) were obtained with remote sensing and airborne measurements (Petzold et al., 2009; Müller et al., 2010; Weinzierl et al., 2011). Recently, synergy analysis methods including ground-based, airborne, remote sensing, and numerical modelling have become important ways to better understand dust evolution (Haarig et al., 2019; Papayannis et al., 2012; Perrone et al., 2004).

Aerosol elastic scattering lidar is widely used for aerosol and cloud research (Killinger and Menyuk, 1987) as it can provide detailed information with high spatial and temporal resolution. However, analysis of this kind of lidar data requires assumptions of lidar ratios and reference values e.g. for retrieving backscatter coefficients (Fernald, 1984; Klett, 1985). The analysis problems can be overcome by High Spectral Resolution lidar (HSRL) (Liu et al., 1999), vibrational Raman lidar (Wandinger, 2005), and pure rotational Raman lidar (Balin et al., 2004). However, the complex configuration of the HSRL

and a pure rotational Raman lidar, as well as the weak scattering intensities of vibrational Raman scattering, impeded the widespread use of these technologies or limit them to night-time measurements. Depolarization lidar measurements have proven to be very useful in detection of non-spherical particles e.g. dust aerosols or ice particles. They can provide the depolarization ratio of particles which can clearly distinguish spherical particles from non-spherical particles (Sassen, 1991). Sun photometers can be used to infer wavelength-dependent optical and microphysical prosperities of aerosols from observing

direct and diffuse solar radiation (Holben et al., 2001; Holben et al., 1998). The ground-based sun photometers aerosol network AERONET (AErosol RObotic NETwork) provides a long-term, continuous, and readily accessible public domain database for aerosol research (Holben et al., 1998).





Various global and regional transport models have been developed and many of them can simulate the transport, transformation, and properties of aerosol particles. Examples of such models are general circulation model ECHAM-

HAMMOZ (Pozzoli et al., 2008b, a), ECHAM/MESSy Atmospheric Chemistry (EMAC) (Roeckner et al., 2006; Jöckel et al., 2006; Jöckel et al., 2010; Kunz et al., 2011), Whole Atmosphere Community Climate Model (WACCM) (Kunz et al., 2011; Smith et al., 2011), Weather Research and Forecasting (WRF) model coupled with Chemistry (WRF/Chem) (Chapman et al., 2008), CONsortium for small-scale MOdeling (COSMO) - its extension Aerosol and Reactive Trace gases (ART) (Vogel et al., 2014), and its successor ICOsahedral Nonhydrostatic (ICON) - its extension Aerosol and Reactive Trace gases (ART)

(Rieger et al., 2015; Weimer et al., 2017)). Special focus has been on mineral dust due to its strong impact on atmospheric radiation forcing (Stocker et al., 2013). A three-dimensional mineral dust model has been developed to study its impact on the radiative balance of the atmosphere (Tegen and Fung, 1994). Mineral aerosol concentration and its total deposition flux over the western North Pacific region were analyzed with the regional chemical transport model (AQPMS) and direct filter samples measurements for the period March 1994 through May 1995 (Uematsu et al., 2003). Recently, the models CAMS (O'Sullivan

et al., 2020), WRF/Chem (Kang et al., 2011) , EMAC (Gläser et al., 2012) , COSMO-ART (Deetz et al., 2016;Vogel et al., 2014) and ICOsahedral Nonhydrostatic model - Aerosols and Reactive Trace gases (ICON-ART) (ICON-ART) (Rieger et al., 2017; Gasch et al., 2017; Hoshyaripour et al., 2019) have been used to predict mineral dust plumes.

For the dust event that occurred in April 2018, we collected a comprehensive set of data and compared it with a global transport model simulation to understand the distribution and evolution of dust near the city of Karlsruhe, in southwest Germany. Two

lidar systems and a sun photometer were used to investigate the dust event employing different retrieval methods. The major objective was to quantify the uncertainties of different measurement and retrieval methods. Furthermore, we wanted to find out how well the dust predictions of the transport model ICON-ART compare with these observations.

This paper is organized as follows. Section 2 describes the remote sensing methods and the model simulations done with ICON-ART. Detailed observation results and properties of the dust plumes are given in Section 3 including a discussion of the

comparison of the different remote sensing methods as well as how they compare to the model predictions.

## 2 Methods

Two lidar systems were used in this study, a vertically pointing system called DWD-DELiRA (LR111-D200, Raymetrics Inc.) and a spatially scanning system called KASCAL (LR111-ESS-D200, Raymetrics Inc.). Both have an emission wavelength of 355 nm and are equipped with elastic, depolarization, and Raman channels, hence providing profiles of extinction coefficients,

backscatter coefficients, and depolarization ratios. Besides, a sun photometer (CE-318, CIMEL, Holben, et al., 1998) provides wavelength-dependent aerosol optical depth (AOD), SSA,  AE, and via inversion the aerosol size distributions (ASD).For



predicting the dust transport and distribution, the online coupled model system ICON-ART (Rieger et al., 2015, 2017) was used. The model system is running in quasi operational mode by Deutscher Wetterdienst (DWD).

## 2.1 Remote sensing instruments

For this observation campaign, the KASCAL and DWD-DELiRA lidar systems were deployed on the campus north of the Karlsruhe Institute of Technology, (8.4298 °E, 49.0953 °N, 119 m above sea level) which are being operated by the institute of meteorology and climate research (IMK) and the Deutscher Wetterdienst (DWD), respectively. The horizontal distance of the two lidar systems was 500 m. The KASCAL lidar system is a mobile scanning system with an emission wavelength of 355 nm. The laser pulse energy and repetition frequency are 32.1 mJ and 20 Hz, respectively. The laser head, 200 mm telescope,

and lidar signal detection units are mounted on a rotating platform allowing zenith angles from -7 ° to 90 ° and azimuth angles from 0 ° to 360º. This KASCAL lidar works automatically, time-controlled, and continuously via a software developed by Raymetrics Inc.. Detailed information can be found at https://www.raymetrics.com/product/3d-scanning-LIDAR, last access: 8 March 2021 (Avdikos, 2015). The fixed vertically pointing lidar (DWD – DELiRA) has the same wavelength, telescope area, and detection channels as the scanning lidar system.

For the data analysis and calibration of the system, we followed the quality standards of the European Aerosol Research Lidar Network (EARLINET) (Freudenthaler, 2016). Extinction and backscatter coefficients at 355 nm were both calculated from the elastic channel using the Klett-Fernald method (Klett, 1985; Fernald, 1984) and are also calculated from the elastic and Raman channels (Ansmann et al., 1992). The extinction coefficients and lidar ratios were also retrieved using a multi-angle method, which is also called ratio method in this paper (Adam, 2012; Gutkowicz-Krusin, 1993). For extinction coefficients

calculated with the Raman method, hamming windows filters whose window length is 300 m (40 bins) were applied to Raman signals, and subsequently the retrievals were done with an average vertical resolution of 150 m. For the backscatter coefficients retrieved based on the Klett-Fernald method, we only show the results for periods free of clouds as the presence of clouds makes it very difficult to choose reasonable reference values for the retrieval methods. We have compared our data analysis algorithm with retrievals by the Single Calculus Chain (SCC) code (EARLINET) and the results are shown in Fig. S3

Multi-angle lidar observations are useful to determine extinction coefficients as mentioned above but can also help to get information of lidar ratios from elastic lidar signals. We found that the backscatter coefficients retrieved using the Klett-Fernald method for different lidar viewing angles showed discrepancies even in a horizontally homogenous atmosphere. These discrepancies varied with lidar ratios assumed. Hence, we can use the backscatter coefficients measured at different lidar viewing angles to determine lidar ratios. A detailed discussion on this is given in the supplement S1.

Particle depolarization is calculated as suggested by (Freudenthaler et al., 2009):

$$\delta^p = \frac{(1 + \delta^m)\delta^v R - (1 + \delta^v)\delta^m}{(1 + \delta^m)R - (1 + \delta^v)} \tag{1}$$

Here $\delta^m$ is the depolarization ratio of gas molecules, $\delta^v$ is the volume depolarization ratio and $R$ is the backscatter ratio:



$$R = \frac{\beta^p + \beta^m}{\beta^m}$$ (2)

Here $\beta^p$ is the backscatter coefficient of particles and $\beta^m$ is the backscatter coefficient of molecules.

On a roof top 25 m above ground level, located between the two lidar systems, a sun photometer (CE-318, CIMEL, Holben, et al., 1998) measures solar radiance at 339, 379, 441, 501, 675, 869, 940, 1021, and 1638 nm. This allows calculation of wavelength-dependent AOD. The sun photometer data can also be used to calculate other aerosol parameters (e.g. SSA, AE, ASD, and complex refractive index (CRI) (Vermeulen et al., 2000; Sinyuk et al., 2020). The SSA is the ratio of scattering coefficient to extinction coefficient, which has a negative correlation with the absorption ability of the aerosol particles. Hence,

this parameter can be used to characterize the scattering and absorption capability of the particles. The AE is a parameter that describes how the optical thickness of an aerosol population depends on the wavelength of the light. Typically, a stronger wavelength dependence occurs when the sizes of particles are smaller than or equivalent to the incident wavelength. Hence, AE has a negative correlation with particle size. From clear sky measurements with the sun photometer, ASD between 0.05 – 15 µm, complex refractive index in the range 1.33 – 1.6 and 0.0005i – 0.5i (Dubovik and King, 2000; Müller et al., 2010) can

be derived.

**2.2 Aerosol Transport Modelling**

For predicting the dust transport and distribution the online-coupled model system ICON-ART (Rieger et al., 2015; Rieger et al., 2017) was used. ICON is a weather and climate model that solves the full three-dimensional non-hydrostatic and compressible Navier–Stokes equations on an icosahedral grid (Zängl et al., 2015). The ART module is an extension of ICON

to include the life cycle and cloud/radiation feedback of aerosols and trace gases. Mineral dust in ART is represented by three lognormal modes with mass median diameters of 1.5, 6.7 and 14.2 µm, and standard deviations of 1.7, 1.6 and 1.5, respectively. The dust emission scheme is based on Vogel et al., (2006) and Rieger et al., (2017) that considers the soil properties (size distribution, residual soil moisture), the soil dispersion state and soil type heterogeneity. The dust removal processes include sedimentation, dry, and wet deposition. The simulations are performed on a global domain including a regional nest (over

North Africa and Europe) with horizontal resolutions of 40 and 20 km, respectively. The vertical resolution of the model ranges from tens of meters to several kilometers from low to high altitudes. At altitudes that often contain the dust layer (2 km – 6 km), the vertical resolution is from 200 to 400 m.

**3 Results and Discussion**

The dust plume was characterised by the methods described above for nearly three days. Spatial-temporal evolution of the dust

plume was characterized by lidar observation and model calculation both of which can provide the vertical structure of dust plume. The backscatter coefficients of dust plumes from scanning lidar (KASCAL), vertical pointing lidar (DWD-DELiRA), and the ICON-ART model simulation are shown in Fig. 1 for April 7th to April 9th, 2018. As can be seen in these three figures,





the dust arrived in Karlsruhe at 11:00 on April 7[th], and lasted about 3 days. Initially, this dust layer showed a maximum in backscatter at an altitude of 2.5 km which subsequently also reached lower altitudes. At 12:00 UTC, April 8[th], another dust

layer which lies between 5.0 -11.0 km arrived at the observation station. Then the dust layer started sinking and overlapped with the lower dust layer at around 3:00 am of April 9[th]. The dust layer heights (vertical extend) and their peak heights (the heights for the maximum backscatter coefficients) for both lidar measurements and ICON-ART prediction are shown in Fig.S2. This figure shows a good agreement in dust layer heights for this two measurements and a statistic analysis shown that the ICON-ART underestimated the peak dust heights on average by 488 ± 865 m. In addition, a vertical profile of backscatter

coefficients for lidar measurements and ICON-ART calculations is shown on the right side of Fig. 1, which also shown a good agreement in dust layer vertical extend but substantial differences in backscatter coefficients. A cloud with a base at 4.5 km appearing at 11:00 (UTC) of April 9[th] made it difficult to retrieve the backscatter coefficients for the aerosol particles below. Hence, the backscatter coefficients of the lidar measurements is not shown for this period

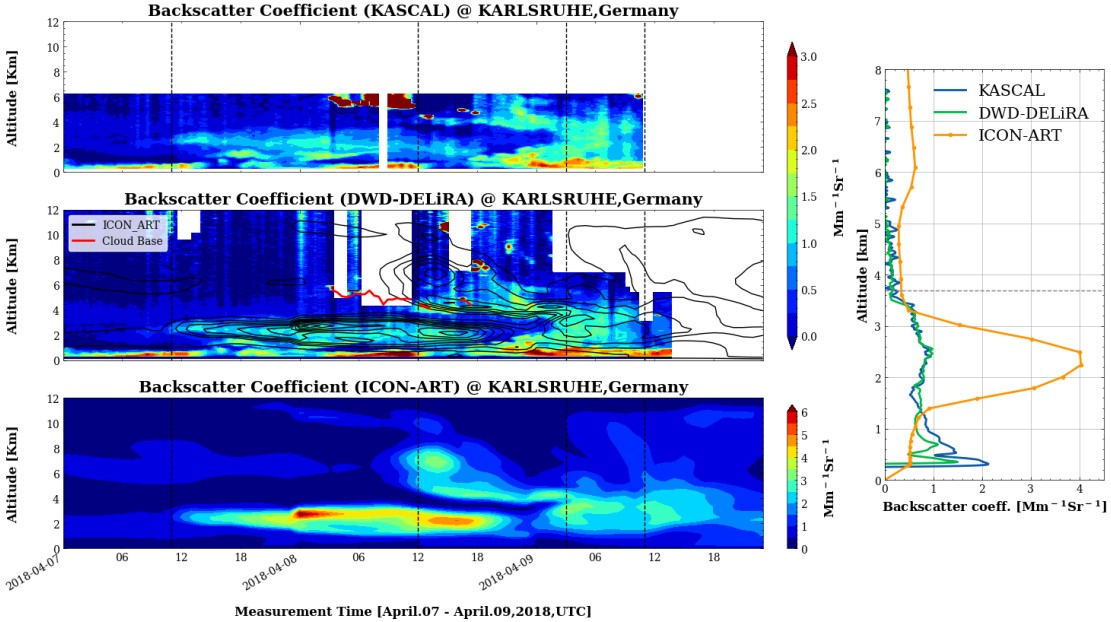


**Figure 1: Time series of backscatter coefficients from KASCAL measurements (upper panel)and from DWD-DELiRA measurements with ICON-ART results shown as black contour lines (middle panel) as well as ICON-ART results (bottom panel) from 7[th] to 9[th], April 2018. Note that the colour scale between measurements and model simulation is different in order to facilitate a visual comparison. The profiles of backscatter coefficients for the DWD-DELiRA lidar and ICON-ART at 23:00 on April 7[th], 2018**

**are shown on the right side of this figure.**

All three panels in Fig. 1 show a good agreement among dust layer height, dust plume arrival times, and dust plume structures. In particular, ICON-ART predicts the arrival time of the dust plume precisely (±20 minutes difference with the observation). This indicated that the model reproduces the synoptic scale processes very well that leads to precise prediction of dust transport. Thus, the general good agreement between lidar measurements and ICON-ART partially validates the model capabilities to





predict dust transport. However, the largest differences between model and measurements are in the absolute values of the backscatter coefficients. The model predicted dust backscatter coefficient values is generally larger than lidar measurement by a factor of 2.2 ± 0.16. Please note that this special model run did only include desert dust. Hence, discrepancies due to boundary layer aerosol particles are expected in this case. Furthermore, there are dust layers predicted by the model for higher altitudes (e.g. a dust plume at around 8 km on April 7th and April 8th) which were not detected by lidar measurements. Potential reasons

for the agreement and differences between lidar observations and model predictions will be discussed in section 3.2.

**3.1 Characteristic properties of the Saharan dust with remote sensing**

During this dust event, the lidars used three optical measurement paths (two vertical measurements and one slant measurement with an elevation angle at 30°). The comparison of these three profiles can be used to test different lidar retrieval methods and to characterize the properties of the dust plume (e.g. horizontal homogeneity of the dust plume). Fig. 2 shows the extinction

and backscatter coefficients obtained for different retrieval methods and different optical paths for the measurement time from 20:21 to 23:54 (UTC) of April 8th, and averaged over 66 minutes for scanning lidar measurements and 213 minutes for vertical lidar measurements. Please note that the scanning lidar measured alternating at the two angles (90° and 30°). A lidar ratio of 55 sr, which is a typical value observed for Saharan dust (Groß et al., 2013), was used in the Klett-Fernald method to retrieve the elastic backscatter coefficients and extinction coefficients. The extinction coefficients and backscatter coefficients

calculated using the above methods as shown in Fig. 2a and Fig. 2c are consistent but the extinction coefficients calculated from the Raman measurements have larger variations. In addition to the classical methods to retrieve extinction coefficients, we also calculated the extinction coefficients from the elastic channels with two elevation angles, which also agrees with the other methods. The denoising methods can have a substantial impact on the remaining variability of the extinction coefficients retrieved from Raman data. In the Fig. S4, we provide extinction coefficients retrieved from Raman data for different filters

and different filter lengths. In addition, the average extinction coefficient and its standard deviation averaging from 4.0 km to 6.0 km altitude is shown in Table S1. From this table, we can find that the mean values of extinction coefficients for different filter type and filter length almost remain constant where the uncertainties vary from around 35 to 5 Mm$^{-1}$ with window length from 82.5 m to 1207.5 m for different types of filters. Hence, the Raman extinction coefficients are affected more by the filter window lengths than the filter type (Shen and Cao, 2019). The backscatter coefficient and extinction coefficient for different

optical paths are shown in Fig. 2b and Fig. 2d, respectively. The consistency of these profiles reflects the high quality of measurements and retrieval algorithms.



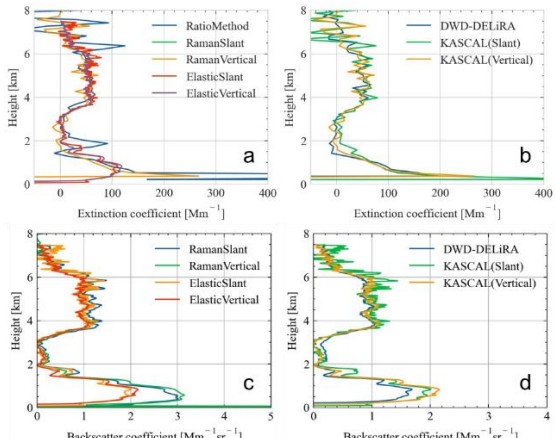

**Figure 2 Extinction coefficients (a) and backscatter coefficients (c) from elastic and Raman methods for different retrieval methods as well as Raman extinction coefficients (b) and elastic backscatter coefficients (d) for different optical paths (vertical from DWD-DELiIRA as well as vertical and slant from KASCAL) from 19:21 to 22:47 (UTC) on 8th, April, 2018 .**

Fig.2c shows that the backscatter coefficients from elastic data for vertical and slant measurement are consistent for Saharan dust particles but inconsistent for boundary layer aerosol particles. This is because a lidar ratio of 55 sr is not suitable for boundary layer aerosol particles at this location (Groß et al., 2017). Therefore, we calculated lidar ratios based on our Raman signals for boundary layer aerosol particles and the Saharan dust particles. The results are shown in the right panel of Fig. 3. The lidar ratio for the dust particles is $46 \pm 5$ sr and for the boundary layer aerosol particles it is $31 \pm 3$ sr for both vertical and slant measurements. We parameterized these lidar ratios being 30 sr and 50 sr respectively as a function of altitude with a single step at 2 km and then used it as lidar ratio for the elastic lidar signal retrieval. The results are shown in the left of Fig. 3. These backscatter coefficients are consistent between vertical and slant measurement for both dust layer and boundary layer. A lidar ratio of 30 sr below 2 km altitude leads to much better agreement of backscatter coefficient profiles from the elastic channel and Raman channel compared to Fig. 2c. However, there remain small differences at low altitudes for backscatter coefficients from elastic data for two elevation angles. This inconsistency may be related to an inhomogeneous atmosphere in the boundary layer as can also be seen in the backscatter coefficients calculated from Raman data.

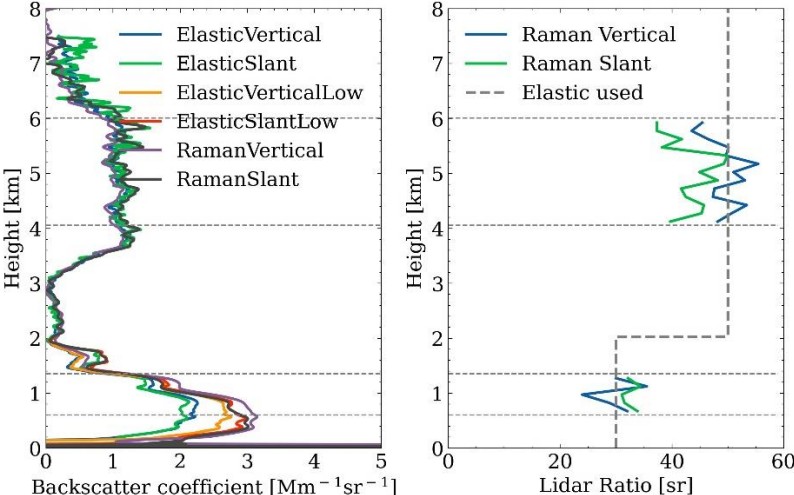


**Figure 3 Backscatter coefficients from elastic and Raman methods for different optical paths (left) and lidar ratios retrieved (right) from 19:21 to 22:47 (UTC) on 8th, April, 2018. The retrieval for the elastic channel data uses two different lidar ratios at different altitudes. (Legend definition is the same as Figure 2. Also, high and low represent in high altitude (above 2km) and low altitude (below 2 km))**

Comparison between the active lidars and the passive sun photometer can help to understand the properties of the dust aerosol particles employing dust aerosol scattering information from different scattering angles to retrieved dust particles' microphysical properties. During this dust event, we have compared the AOD from vertical lidar measurements and a sun photometer for two continuous days (April 7th-8th). The AOD from the sun photometer is the AERONET version 3 level 2.0 product (Sinyuk et al., 2020) while that of lidar measurement was corrected in following ways. Firstly, as discussed above,

two aerosol layers existed with different lidar ratios. Hence, we used two different lidar ratios at different altitudes to retrieve the backscatter coefficients, which are shown in Fig. S5. We have used a lidar ratio of 55 sr for the upper layer (above 2 km, red line) and 30 sr for the lower layer (below 2 km, green line), typical values for Saharan dust and boundary layer aerosol (Groß et al., 2013). Secondly, constant backscatter coefficients are used in the lidar overlap region and these constant values are set to be the backscatter coefficient at 255 m (the overlap region of DWD-DELiRA). Finally, the AOD in the far range

(e.g. stratosphere) is assumed to be zero. The hourly AODs from the sun photometer, the vertical lidar (DWD-DELiRA) and the ICON-ART model are shown in Fig. 4. All these three methods show a similar trend with AODs increasing from around 0.13 to 0.45 during these two days. However, the average AOD retrieved from the lidar data for two days is systematically lower by $0.037 \pm 0.02$ than that from the sun photometer after wavelength conversion to 340 nm assuming an AE of one. The average stratospheric AOD of for the years 2018-2019 in Northern Hemisphere was 0.01 at 340 nm (Kloss et al., 2020). Hence,

the averaged AOD measured by the sun photometer is still larger by $0.027 \pm 0.02$ than the AOD from the lidar measurement even considering the stratospheric AOD. This bias may due to an inappropriate assumption of constant backscatter coefficients in the overlap region of the lidar. Such an uncertainty of AOD corresponds to an uncertainty in backscatter coefficients of 3.5





± 2.6 Mm⁻¹ sr⁻¹ in the overlap region. On April 8ᵗʰ, clouds lead to increased uncertainties in AOD retrievals from the lidar measurements. Hence, the AOD from lidar measurements is given only for clear sky periods.


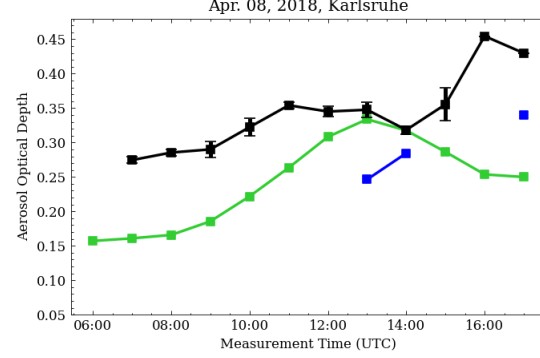

**Figure 4: AOD from lidars (blue circles) and sun photometer (black squares) on 7ᵗʰ and 8ᵗʰ of April for 1 hour temporal resolution.**

For this dust event, vertical and slant volume ($\delta^v$) and particle ($\delta^p$) depolarization ratios were measured by the two different lidar systems, and the volume depolarization ratios for these two elevation angles are shown in Fig. 5. No obvious difference
between vertical and slant measurements was found for volume depolarization ratios and particle depolarization ratios. This may mean that the dust particle had no specific orientation. The particle depolarization ratio of this dust plume was $0.33 \pm 0.07$ which is a typical value for Saharan dust particles (Freudenthaler et al., 2009).

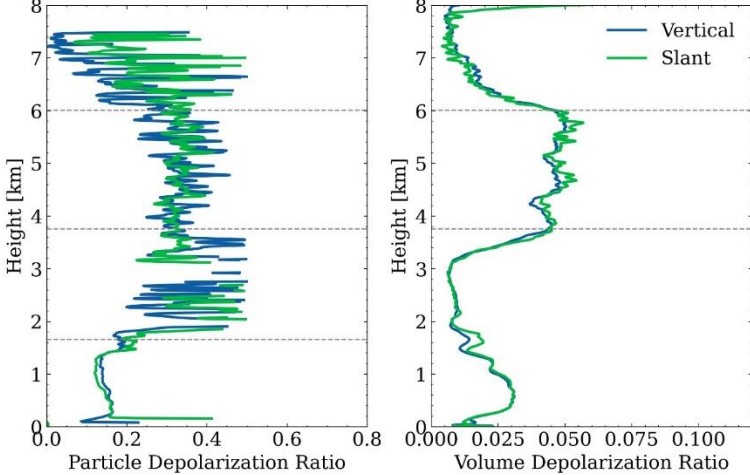

**Figure 5: Volume and particle depolarization ratio in vertical and slant observation direction from the scanning lidar (KASCAL) from 19:21 to 22:47 (UTC) on 8th, April, 2018.**





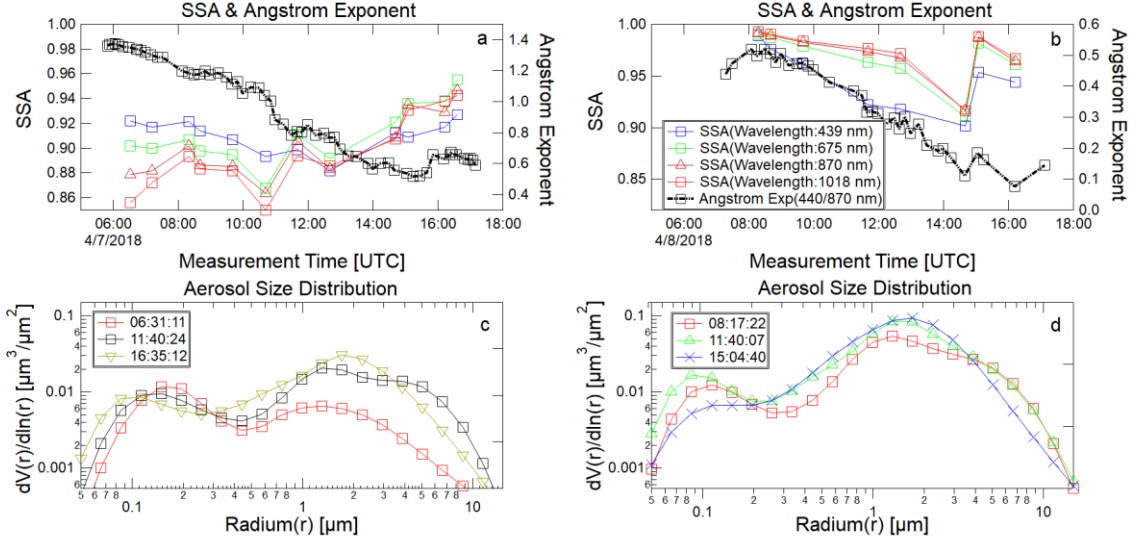

**Figure 6: Single scattering albedo and Angstrom exponents (a, b), and aerosol size distribution (c, d) from sun photometer**
**measurements for April 7th and 8th, 2018.**

Fig. 6 shows the SSA, AE, and) ASD calculated based on the sun photometer measurements on April 7th and 8th. A decreasing AE from 1.38 to 0.08 at wavelengths of 440/880 nm shown in Fig. 6a and Fig. 6b corresponds to a smaller wavelength dependence of AOD, which may be caused by larger particles. Particle size distributions provided by sun photometer retrievals are shown in Fig. 6c and Fig. 6d. They indeed show increasing amounts of larger particles. The maximum average volume
concentration of coarse mode increased from around 0.007 $\mu m^3/ \mu m^2$ in the early morning of April 7th (before dust arrival) to 0.093 $\mu m^3/ \mu m^2$ in the afternoon of April 8th. Besides, the measured SSA values 0.88 – 0.92 (439 nm), 0.87-0.96 (675 nm), 0.86-0.94 (870 nm) , 0.85-0.94 (1018 nm) agree with previous observations of 0.96 (537 nm), 0.98 (637 nm) (Schladitz et al., 2009), 0.91 (450 nm), 0.96 (550 nm), 0.98 (950 nm) (Müller et al., 2011), and 0.96 – 0.99 (530 nm, D < 3 $\mu$m, airborne measurements) (Petzold et al., 2011). The SSAs measured  for this case and in previous studies are listed in table 1.


**Table1**. Overview of SSAs measured for Saharan dust

| SSA | Wavelength [nm] | Reference |
|---|---|---|
| 0.88 – 0.92 | 439 | This work |
| 0.91 | 450 | Müller et al., 2011 |
| 0.96 – 0.99 | 530 | Petzold et al., 2011 |
| 0.96 | 537 | Schladitz et al., 2009 |
| 0.96 | 550 | Müller et al., 2011 |
| 0.98 | 637 | Schladitz et al., 2009 |
| 0.87-0.96 | 675 | This work |



| 0.86-0.94 | 870 | This work |
|-----------|-----|-----------|
| 0.98 | 950 | Müller et al., 2011 |
| 0.85-0.94 | 1018 | This work |

## 3.2 Regional model predictions and comparison with observations

Model simulations and lidar observations were used to study the spatial and temporal evolution of a dust plume in this study.
The comparison between model and lidar result can be used to evaluate the performance of model simulation including dust
layer height, dust arrival time, dust structure, and dust optical parameters. The evolution of the dust plume over Karlsruhe
predicted by the ICON-ART model is shown in the lower panel of Fig. 1. According to the model simulation, the dust layer
arrived in Karlsruhe at 11:00 of April 7th and this plume passed over that location for nearly two and a half days. Two dust
layers were observed from time 12:00 (UTC) of April 8th to the morning of April 9th then they merged. Comparison between
model prediction and lidar measurement is shown in the middle panel of Fig. 1, where the black contour line is the modelled
backscatter coefficient and the contour fill is the lidar (DWD - DELiRA) observation. The red thick line is the cloud base
height from the lidar measurement. The comparison shows that dust plume arrival time, layer height, and dust structure are
consistent between lidar measurement and model simulation. Although the lidar data shows more details of the dust plume
structures, the agreement with the model is surprisingly good. On the other hand, the lidar has difficulties to measure e.g. thin
layers of dust especially in the presence of clouds. Therefore, a comparison for thin dust layers is not always meaningful. In
terms of backscatter coefficients, the results from ICON-ART predictions is a factor of $2.2 \pm 0.16$ lager than that from the lidar
measurement as can be seen by comparing the color bar of the contour fill of Fig. 1. Besides, the AODs for three wavelengths
from sun photometer and model calculation are shown in Fig. 7. This figure shows that, the AODs from sun photometer and
model show a similar trend. However, the modeled AODs are systematically lower as they only reflect the dust aerosol. Before
dust arrived (11:00 UTC), the AODs from the model are larger than that measured by the sun photometer at 1020 nm, which
may be due to a limited sensitivity of sun photometer as a few small particles cannot be detected in the infrared regions. After
arrival of the dust plume, the AOD from model calculation is systematically lower than the sun photometer measurement and
the bias between model and sun photometer increase with the wavelength towards the ultraviolet (UV) spectral region. In other
word, the discrepancy is wavelength dependent with a bigger difference in the UV. The phenomenon is attributed to an
underestimation of small particles in the model calculation (Hoshyaripour et al., 2019). In addition, the AOD from ICON-ART
is lower than the sun photometer results although the backscatter coefficients are twice the lidar data. This might be explained
by the non-sphericity of the dust particles, which are assumed to be spherical in the model (Hoshyaripour et al., 2019).





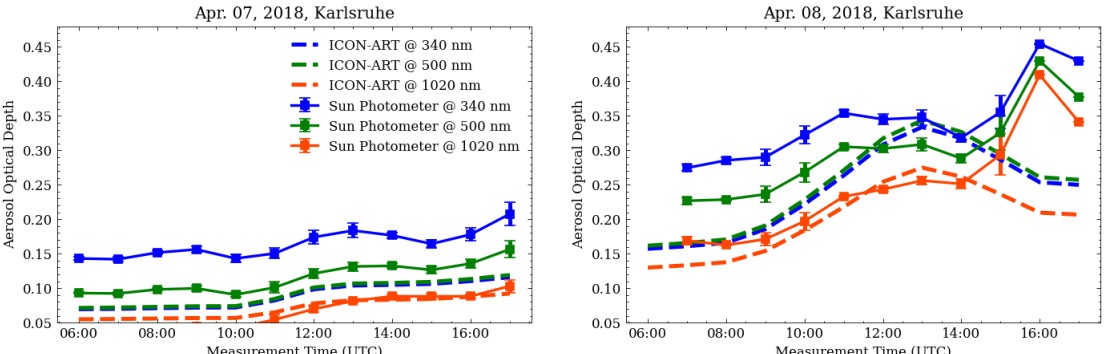

**Figure 7: AOD from the sun photometer and ICON-ART model simulation on 7th and 8th of April for 1hour temporal resolution.**

## 4 Conclusions

The evolution and the properties of a Saharan dust plume were characterized for two and a half days combining a scanning lidar, a vertical lidar, a sun photometer, and a regional transport model. The comprehensive dataset from different methods could characterize the dust plume in different ways, thus providing additional information for further analysis.

The comparison of extinction and backscatter coefficients for different retrieval methods was used to quantify uncertainties of the different methods and the impact of different denoise filters on extinction coefficients from Raman scattering lidar signals. The consistency among three different lidar laser beam paths reflects the high quality of the measurements as well as the retrieval algorithms. Vertical and slant volume and particle depolarization ratio are allowed to study the non-spherical and orientation of dust particles. Comparison between lidar and sun photometer measurements has proven useful to study the dust optical properties like aerosol optical depth and to get information about lidar parameters like the lidar ratio. Wavelength-dependent optical parameters and microphysics of dust particles provided by the sun photometer indicated larger particles over the observation station. Comparison between lidar measurements, sun photometer and ICON-ART predictions shows a good agreement for dust arrival time, dust layer height, and dust structure but also that the model overestimates the backscatter coefficients by a factor of $2.2 \pm 0.16$ and underestimate aerosol optical depth by a factor of $1.5 \pm 0.11$.

**Code availability:** The code used to analyse the lidar data is property of Raymetrics Inc, but we have shown that it gives the same results as the code single calculus chain (SCC) provided by EARLIENT (https://www.earlinet.org/index.php?id=earlinet_homepage, last access: 8 March 2021) and public available. The ICON-ART code is license protected and can be accessed upon request to Heike Vogel (heike.vogel@kit.edu).

**Data availability:** The lidar raw data are available via the open access data repository KIT open. Sun photometer are available from the AErosol RObotic NETwork (Aeronet) data center at https://aeronet.gsfc.nasa.gov/ (last access: 08 May



2021). ICON-ART model simulation results are available upon request from the data originator (DWD; datenservice@dwd.de).


**Author contribution:** FW and HS performed the measurements. GAH, HV, VB, and JF conducted the model simulations and post processing of the outputs. HZ and FW analysed the remote sensing data. HZ did write the manuscript with support from FW and HS as well as contributions from all co-authors.


**Competing interests:** The authors declare that they have no conflict of interest.

**Acknowledgements:** Support by the staff of the Institute of Meteorology and Climate Research, financial support by the project Modular Observation Solutions for Earth Systems (MOSES) of the Helmholtz Association (HGF) and the German
Federal Ministry for Economic Affairs and Energy project PerduS (support ID: 0325932B) are gratefully acknowledged.

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
