# Peer review of "Investigation of a Saharan dust plume in Western Europe by remote sensing and transport modelling"

_Atmospheric Measurement Techniques, 2021_

## Author Comment (AC1)

**'Comment on amt-2021-193', by Aristeidis Georgoulias, 04 Aug 202**

Dear authors,

Congratulations on this comprehensive study. Such studies are useful in order to bring out the potential of using different observational datasets and modeling systems to better characterize episodic events over background areas.

I strongly urge you to enrich your reference list and link your study with previous studies in the area combining satellite, model, and observational data. Two such studies are:

Akritidis, D., Katragkou, E., Georgoulias, A. K., Zanis, P., Kartsios, S., Flemming, J., Inness, A., Douros, J., and Eskes, H.: A complex aerosol transport event over Europe during the 2017 Storm Ophelia in CAMS forecast systems: analysis and evaluation, Atmos. Chem. Phys., 20, 13557–13578, https://doi.org/10.5194/acp-20-13557-2020, 2020.

Osborne, M., Malavelle, F. F., Adam, M., Buxmann, J., Sugier, J., Marenco, F., and Haywood, J.: Saharan dust and biomass burning aerosols during ex-hurricane Ophelia: observations from the new UK lidar and sun-photometer network, Atmos. Chem. Phys., 19, 3557–3578, https://doi.org/10.5194/acp-19-3557-2019, 2019.

Specifically, the study from Akritidis et al. (2020) highlights the ability of CAMS to capture the complex aerosol transport event (dust and smoke) over central-western Europe during Storm Ophelia while the second study is mostly ground-based.

R: Thank you for pointing us to these interesting studies, which we included in the introduction.

Best regards,

Hengheng Zhang and all co-authors

---

## Author Comment (AC2)

**Referee #1 comments:**

We thank the referee for the useful comments, which helped us to improve the quality of our manuscript.

In the following, the referees' comments are given in black.

Our point-to-point replies are marked by "R" and are in blue.

Changes to the manuscript text are in green.

**General remarks:**

The study deals with the description of a long-range Saharan dust plume that affected the Central Europe in April 2018 and captured by ground-based instruments (lidars, sunphotometer) operating at Kalrsruhe (Germany). Moreover, an evaluation of the ICON-ART transport model is performed. I think that several modifications on the manuscript are needed in order to be acceptable for publication in AMT. For instance, it has not been clear by the authors which is the added value of the current study with respect to previous similar analyses. Likewise, an intercomparison of the obtained findings with those reported in past studies is missing. A critical point which must be clear to the reader is to highlight the purpose of the current study. There are parts in which different retrieval methods (raman vs klett) are compared, different observational geometries (vertical point vs off-zenith vertical profiles) are discussed, different remote sensing techniques (active vs passive) are employed and dust numerical simulations are evaluated against ground-based measurements. But it is not clear what is the exact proposition from this exercise (e.g., to deploy similar instrumentation for desert dust studies?). Even though the amount of data/techniques sounds impressive, the way that they are presented is confusing to my opinion. As you will see in my comments below it is required a restructure of the paper sections. Finally, please consider to improve the English writing throughout the manuscript.

R: It is our aim to demonstrate how useful multiple angle lidar measurements can be in addition to vertical lidar and sun photometer data and what kind of understanding of the aerosol properties can be achieved by combining the different measurement techniques. In addition, we want to understand the quality of the dust plume predictions with one of the state of the art transport models, the ICON-ART model, by comparison with these observational data. To point this out, we have added information about similar previous studies to the introduction and modified its final part as follows:

"The major objective is to quantify the uncertainties of different measurement and retrieval methods including a demonstration how useful multi angle lidar measurements can be in addition to vertical lidar and sun photometer data and what kind of understanding of the aerosol properties can be achieved by combining the different measurement techniques. Furthermore, we want to understand the quality of the dust plume predictions with one of the state of the art transport models, the ICON-ART model, by comparison with these observational data. This paper is organized as follows. Section 2 describes the remote sensing methods and the model simulations done with ICON-ART. Details of our dust observations and its properties are discussed in section 3 including a comparison of the different remote sensing methods as well as how they compare to the model predictions. In the final section we provide some conclusions from this."

**Minor edits:**

1. Lines 12-13: Provide the wavelength

R: The wavelength of our lidar is 355 nm, and we have added this information to the abstract of the manuscript.

The related text has changed into:

"The lidar measurements at a wavelength of 355 nm show that the dust particles had backscatter coefficients of  $0.86 \pm 0.14 \text{ Mm}^{-1} \text{ sr}^{-1}$ , an extinction coefficient of  $40 \pm 0.8 \text{ Mm}^{-1}$ , a lidar ratio of  $46 \pm 5$  sr, and a linear particle depolarization ratio of  $0.33 \pm 0.07$ ."

2. Lines 34-35: Could you please explain better this sentence? Which are the problems for CALIOP to depict the vertical structure of dust layers?

R: We think it is fair to say that their capabilities to retrieve dust plume structures is limited especially compared to ground based active remote sensing methods.

"However, their data still has limitations compared to ground based active remote sensing methods e.g. concerning characterisation of the structures of dust plumes especially for low aerosol particle concentrations (Ma et al., 2018)."

- Line 47: Replace "Recently, synergy analysis methods..." with "Recently, synergistic approaches/methods...".
   R: We have changed this in the revised manuscript.
- 4. Lines 54-56: Rephrase and explain better this sentence.
  - R: We have changed this sentence into:

"However, the complex configuration of some of these systems, as well as the relatively weak intensities of vibrational Raman scattering, impeded the widespread use of these technologies or limit them to night-time measurements."

5. Lines 74-77: Check also the SDS-WAS in which several regional models provide short-term dust forecasts over the NAMEE domain.

R: We agree that it is useful to add information on the SDS-WAS program at this point and modified the text as follows: "Various models like CAMS (O'Sullivan et al., 2020), WRF/Chem (Kang et al., 2011), EMAC (Gläser et al., 2012), COSMO-ART (Deetz et al., 2016;Vogel et al., 75 2014) and ICOsahedral Nonhydrostatic model - Aerosols and Reactive Trace gases (ICON-ART) (Rieger et al., 2017; Gasch et al., 2017; Hoshyaripour et al., 2019) have been used to predict mineral dust plumes. A systematic validation of ICON-ART is beyond the scope of this study and has been done in previous works (Rieger et al., 2017; Gasch et al., 2017; Hoshyaripour et al., 2019). However, a multi model forecast comparison is beyond the scope of this study but is available by the Sand and Dust Storm Warning Advisory and Assessment System (SDS-WAS) (https://sds-was.aemet.es/forecast-products/dust-forecasts/forecast-comparison, last access September 22, 2021), a program of the World Meteorological Organisation (WMO)"

- 6. Lines 90-91: Not only ASD but SSA is also retrieved. Please make the appropriate corrections in this sentence.
  R: We have modified this sentence as follows:
  "Besides, a sun photometer (CE-318, CIMEL, Holben, et al., 1998) provides wavelength-dependent aerosol optical depth (AOD), AE, and via inversion the aerosol size distributions (ASD), and SSA."
- 7. Line 114: It is strange that for the first time in the manuscript you are referring to Figure S3. Also it is missing a short description about this comparison.

R: We agree that it is strange to show Fig.S3 first in the manuscripts. And we also think that the comparison between our data analysis procedure and Single Calculus Chain (SCC) code (EARLINET) is necessary if we want to use our own data analysis procedure. All these things are obvious, so we think Fig. S2 did not bring too much information in the manuscripts. Therefore, we would like to delete Fig S2 and related description.

8. Results and discussion: It would be useful to add a section describing the factors driving the emission and transport of the Saharan dust plume towards central Europe. Such analysis should include model outputs (e.g., meteorology, dust) as well as ground-based observations (these have been already provided but not in an appropriate place) and satellite retrievals thus providing a complete overview.

R: The factors driving the emission and transport of the Saharan dust plume towards central Europe are of course of great interest but not really in the focus of our study which is discussing different ground based remote sensing methods and how they compare with one transport model as pointed out above. However, we have added some more information on the main factors relevant for emission and transport of Saharan dust towards central Europe and discuss them in the context of our comparison of model results with observations in Karlsruhe in section 3 as they may have some influence on the model performance. The added sentence is as follow:

"In early April 2018, a far southward reaching upper-level trough associated with a large low-pressure complex in the western North Atlantic lad to a cold front with strong surface winds and dust emission in the Northern Sahara in Morocco and Algeria. The dust was transported northward into the western Mediterranean where it entered a warm conveyor belt that effectively lifted the dust and transported it towards central Europe. "

9. Figure 1:

Which is the off-zenith angle for the KASCAL aerosol profiles?

R: We have added the off-zenith angle for KASCAL aerosols profiles in the right figure. We did measured with KASCAL  $0^{\circ}$ ,  $60^{\circ}$  off zenith angle ( $90^{\circ}$  and  $30^{\circ}$  elevation angle) and we add this information in the manuscript .

"The scanning lidar was operated doing vertical and slant measurements at 90° and 30° elevation angle alternatingly with integration times for each observation angle of 300 s."

Use common colorbar for the three curtain plots in order to facilitate a visual intercomparison among them.

R: We have changed the colorbar of Figure 1 into a common colorbar.

It would be interesting to make a quantitative comparison (e.g. bias) between the curtain plots. To realize, you have to regrid the altitude-time plots and project them in a common grid.

R: For this purpose we have included the ICON-ART data in the middle panel as contour lines over the lidar data demonstrating that the model predicts the dust plume arrival time and structure quite well. For a quantitative comparison we have plotted the backscatter coefficients for one time interval separately. We think that this allows a good comparison of the backscatter coefficients.

I suggest to remove the black curves from the middle plot. I don't see why they are useful and in some cases it is hard to distinguish them (packing). Moreover, the labels are missing.

R: We have reduced the line thickness of the black curve representing the model result in the middle plot to improve the visibility. We consider it useful to demonstrate the agreement regarding dust plume arrival time and structure.

How you have selected the timeframe for the backscatter plot (right figure)?

R: We choose a period without clouds and a distinct dust layer for this comparison. We indicate this period in Figure 1 now. The modified Figure 1 and caption are given below:

Figure 1: Time series of backscatter coefficients from KASCAL measurements (a) and from DWD-DELiRA measurements with ICON-ART results shown as black contour lines (b) as well as ICON-ART backscatter coefficients (c) and linear volume depolarization ratios from KASCAL measurements (d) from April 7th to 9th, 2018. Please note that the model data only includes the Saharan dust while the lidar data shows also other aerosol particles and clouds. The profiles of backscatter coefficients measured by the two lidars from 22:30 to 23:30 and predicted by ICON-ART for 23:00 on April 7th, 2018 (indicated as C1 in the contour plots) are shown on the right side of this figure. The vertical dashed lines in the

contour plots indicate dust arrival (T1), second dust layer appeared (T2), and the two dust layers merged (T3). C1 and C2 represent time periods used for a more detailed data analysis.

How the backscatter coefficient by the model has been calculated?

R: Mineral dust in ICON-ART is represented by three lognormal modes with mass median diameters of 1.5, 6.7 and 14.2  $\mu$ m, and standard deviations of 1.7, 1.6 and 1.5, respectively. The output of the ICON-ART model is mass concentrations of these three modes. The mass backscatter cross section (m2/g) is provided by Meng et al. (2010). We have added this information in section 2.2 of the revised manuscript as follows.

"For this study, the altitude dependent backscatter coefficients and column AODs were used to compare ICON-ART calculations to the results from lidar and sun photometer measurements based on the three lognormal dust size distributions in ICON-ART and the mass backscatter cross sections, mass extinction cross sections provided by Meng et al. (2010)."

Why the modelled backscatter coefficient is so much overestimated?

R: Besides potential overestimation in particle mass or size, we consider a main reason the assumption that the dust particles are spherical to calculate the backscatter coefficients from the dust mass size distributions. As we discussion section 3.2, the AOD from model calculation is systematically lower than from sun photometer measurements as their bias is wavelength dependent and increased with decreasing wavelength towards the UV regions. This is potentially related to an underestimation of the number of small particles as shown by Hoshyaripour et al., (2019). Therefore, the overestimation of backscatter coefficients is most likely not due to overestimated particle numbers.

Fig. 2 of Hoshyaripour et al., (2019) shows the scattering phase function at wavelengths of 532 nm and 1064 nm for spherical (SPH) and non-spherical (NSP) particles for the three lognormal modes. The spherical particles have larger backscatter coefficients (at 180°) than non-spherical particles. This is the reason that the backscatter coefficients are so much overestimated. The physical meaning behind is phenomenon is that for spherical particles also surface waves can contribute to the backscatter, hence causing larger backscatter coefficients for spherical particles (Hovenac and Lock, 1992).

In the revised manuscript, we show now the mass backscatter coefficient for non-spherical particle parameters as provided by Meng et al. (2010). The revised modelled backscatter coefficients can be seen in the revised Fig.1. After using parameters for non-spherical particles, the predicted backscatter coefficient is generally on average larger by only a factor of  $1.01 \pm 0.56$  compared to the lidar measurements at wavelength of 355 nm.

---

## Author Comment (AC3)

**Referee #2 comments:**

We thank the referee for the useful comments, which helped us to improve the quality of our manuscript.

In the following, the referees' comments are given in black.

Our point-to-point replies are marked by "R" and are in blue.

Changes to the manuscript text are in green.

**Referee #2 comments:**

**General remarks:**

An intense Saharan dust outbreak was observed with two lidar systems and a sun photometer. The observations are compared to model prediction by ICON-ART model. Arrival time and dust layer heights agreed for the first dust plume, for the second dust plume at higher altitudes the agreement is less good. The backscatter coefficient was overestimated by the model, the AOD underestimated. A lot of work was put into the lidar analysis for one vertical pointing and one slanted (30°) and vertical pointing lidar system. The structure and the language needs improvements.

Comparing a single dust event with the model predictions of a single model is surely a lot of effort, but it is not state of the art anymore. Therefore, major revisions are necessary before publication. I would consider a single dust event evaluated with different dust transport models more interesting for the community. Doing so, the strengths and weaknesses of the models could be pointed out. At least two or three more dust transport models should be compared.

Or evaluate multiple dust events with the same model to get some statistics when the model predictions are in line with the observations and when not and to look for the reasons. One event evaluated with one model is surely not enough for a publication in 2021.

Already much more complex publications concerning the comparison of ground-based remote sensing (lidar) and dust transport models are present in literature, e.g. for a dust plume across the Atlantic Ocean (e.g., Kanitz et al., 2014), extreme dust events (e.g., Solomos et al., 2017), year-long statistics (e.g., Mona et al., 2014), multi-station statistics (e.g., Soupiona et al., 2020) and fine and coarse dust mass concentrations (e.g., Ansmann et al., 2017). It seems that the present study was performed without the knowledge of the progress made in the past decade.

The list of dust transport models mentioned in the introduction is not complete and should be updated. NMMB/BSC-Dust and SKIRON came immediately into my mind, but certainly there are more.

R: We are aware of previous comparisons of dust transport models and ground-based observations but it is not the aim of this work to provide a systematic evaluation of models. Our objective is to highlight the potential of multiple angle lidar measurement data in detailed validation of model outputs beyond optical depth and concentration. To do so, we need to know the underlying assumption and parameterization in the model system like dust optical properties. Such information are not simply provided alongside the forecast data. Instead, one need to work closely with the model developers closely to make sure that the comparisons are consistent and robust.

Currently 12 models are compared in an operational way through the WMO SDS-WAS (Sand and Dust Storm Warning Advisory and Assessment System) project. We added the reference to WMO SDS-WAS in the introduction. In additional, we have added some literature related to our work in introduction section to make our manuscript more complete. The added text is follows:

"Various studies characterized Saharan dust either near the sources as well as during and after long range transport. Freudenthaler et al. (2009) reported pure Saharan dust depolarization ratio profiling at several wavelengths during the Saharan

Mineral Dust Experiment (SAMUM) 2006. Kanitz et al. (2014) observed Saharan dust with shipborne lidar from 60° to 20°W along 14.5°N. Soupiona et al. (2020) studied dust properties and its impact on radiative forcing over the northern Mediterranean region based on EARLINET observations. The three-dimensional evolution of Saharan dust transport toward Europe was studied based on a 9-year EARLINET-optimized CALIPSO dataset (Marinou et al., 2017). The Copernicus Atmosphere Monitoring Service (CAMS) forecast systems simulated the aerosol transport events over the Europe during the 2017 storm Ophelia and validated these results with passive (MODIS: Moderate Resolution Imaging Spectroradiometer aboard Terra and Aqua) and active (CALIOP/CALIPSO: Cloud-Aerosol LIdar with Orthogonal Polarization aboard Cloud-Aerosol Lidar and Infrared Pathfinder Satellite Observations) satellite sensors as well as ground-based measurements (EMEP: European Monitoring and Evaluation Programme). Osborne et al. (2019) compared model simulations with ground based remote sensing measurements (lidar and sunphotometer network). A comparison of dust observations by lidar and BSC-DREAM8b model results was studied by Mona et al. (2014). "

Literature is full of comparisons of models with observations. A more careful literature research is definitely necessary to place your observations in a broader context and to clearly state the novelty of your study.

R. We agree that there are various reports on different dust events in the literature. Many of them analysed the dust plumes also with advanced lidar systems e.g. employing multiple wavelengths and combinations of different remote sensing instruments. Furthermore, there are several attempts to improve, compare, and validate models predicting dust plume generation and transport. In contrast to these studies, we wanted to show potential advantages of multiple angle lidar measurements in terms of characterising dust particles especially concerning optical properties and for independently determining the lidar ratios. Furthermore, we wanted to use our observational data to compare the results with the ICON-ART model outcome. We wanted to investigate which parameters can be compared also to optimize our observations for future more systematic field measurements then allowing for evaluating multiple dust events with the same model. Therefore, our study is focused on the scanning lidar contributions to improve dust plume observations and a first comparison with the ICON-ART model. We think that the manuscript contains valuable results on the scanning lidar contributions to characterize dust plumes as well as the potential of the ICON-ART model for prediction of Saharan dust plumes which reward publication. We have added corresponding references to the introduction to clarify the objectives of this study compared to previous ones.

**Major comments:**

1. The range limitations of KASCAL (Fig. 1) were not discussed. No data were reported above 6 km. Why?

R: For this study, the scanning lidar was measuring alternating at two different observation elevation angles of 90° & 30°
leading to shorter integration times compared to the continuously vertically pointing lidar. Due to the shorter integration
time of 600 s and the existence of clouds for some periods, the quality of the KASCAL data is relatively low for the 6-8
km altitude range. However, in order to show these contour figures consistently, we replot this figure and show the data
below 8 km for all measurements and model calculations. The modified figure 1 is shown below:

Figure 1: Time series of backscatter coefficients from KASCAL measurements (a) and from DWD-DELiRA measurements with ICON-ART results shown as black contour lines (b) as well as ICON-ART backscatter coefficients (c) and linear volume depolarization ratios from KASCAL measurements (d) from April 7th to 9th, 2018. Please note that the model data only includes the Saharan dust while the lidar data shows also other aerosol particles and clouds. The profiles of backscatter coefficients measured by the two lidars from 22:30 to 23:30 and predicted by ICON-ART for 23:00 on April 7th, 2018 (indicated as C1 in the contour plots) are shown on the right side of this figure. The vertical dashed lines in the contour plots indicate dust arrival (T1), second dust layer appeared (T2), and the two dust layers merged (T3). C1 and C2 represent time periods used for a more detailed data analysis.

2. Section 3, lines 149-180: The backscatter coefficient itself does not tell you, which layer is dust and which not. Throughout the section, you are writing "dust". At this point, you haven't shown the depolarization ratio yet to demonstrate, that your measured backscatter coefficient is really dust.

R: We agree that the backscatter coefficients alone do not tell us which layer is dust. Therefore, we have reformulated the beginning of section 3 as follows and added the particle depolarization to figure 1.

"The corresponding backscatter coefficients from the scanning lidar (a), the vertical pointing lidar (b), and the ICON-ART model simulation (c) together with the linear depolarization values of the KASCAL (d) for April 7th to April 9th, 2018 are shown in the Fig. 1"

Line 171: This statement is true for the first dust layer arriving on 7 April 2018. However, the second dust layer arriving in the evening of 9 April 2018 at around 5-6 km height was predicted by the model around 12 hours too early.
 R: The detection of the dust layer predicted to arrive around 11:00 on April 9th, 2018 at an altitude of 5-7 km was impeded by the presence of clouds as indicated by the cloud base in the middle panel of Fig. 1. As shown in Fig. S4, we could

detect the dust plume arriving at the same time as predicted but only below the clouds. However, we have to admit that this is obviously not an ideal example. The Fig.S2 has changed into Fig. S4 in the revised supplement.